# In Vitro Activity of Cefiderocol on Multiresistant Bacterial Strains and Genomic Analysis of Two Cefiderocol Resistant Strains

**DOI:** 10.3390/antibiotics12040785

**Published:** 2023-04-20

**Authors:** Michela Padovani, Anna Bertelli, Silvia Corbellini, Giorgio Piccinelli, Francesca Gurrieri, Maria Antonia De Francesco

**Affiliations:** Institute of Microbiology, Department of Molecular and Translational Medicine, University of Brescia-ASST Spedali Civili, 25123 Brescia, Italya.bertelli009@unibs.it (A.B.);

**Keywords:** resistance, cefiderocol, multi-drug resistance (MDR), whole-genome sequencing (WGS), iron transporters

## Abstract

Cefiderocol is a new siderophore cephalosporin that is effective against multidrug-resistant Gram-negative bacteria, including carbapenem-resistant strains. The aim of this study was to evaluate the activity of this new antimicrobial agent against a collection of pathogens using broth microdilution assays and to analyze the possible mechanism of cefiderocol resistance in two resistant *Klebsiella pneumoniae* isolates. One hundred and ten isolates were tested, comprising 67 *Enterobacterales*, two *Acinetobacter baumannii*, one *Achromobacter xylosoxidans*, 33 *Pseudomonas aeruginosa* and seven *Stenotrophomonas maltophilia.* Cefiderocol showed good in vitro activity, with an MIC < 2 μg/mL, and was able to inhibit 94% of the tested isolates. We observed a resistance rate of 6%. The resistant isolates consisted of six *Klebsiella pneumoniae* and one *Escherichia coli*, leading to a resistance rate of 10.4% among the Enterobacterales. Whole-genome sequencing analysis was performed on two cefiderocol-resistant *Klebsiella pneumoniae* isolates to investigate the possible mutations responsible for the observed resistance. Both strains belonged to ST383 and harbored different resistant and virulence genes. The analysis of genes involved in iron uptake and transport showed the presence of different mutations located in *fhu*A, *fep*A, *iut*A, *cir*A, *sit*C, *apb*C, *fep*G, *fep*C, *fet*B, *yic*I, *yic*J, and *yic*L. Furthermore, for the first time, to the best of our knowledge, we described two *Klebsiella pneumoniae* isolates that synthesize a truncated fecA protein due to the transition from G to A, leading to a premature stop codon in the amino acid position 569, and a TonB protein carrying a 4-amino acid insertion (PKPK) after Lysine 103. In conclusion, our data show that cefiderocol is an effective drug against multidrug-resistant Gram-negative bacteria. However, the higher resistance rate observed in *Enterobacterales* underlines the need for active surveillance to limit the spread of these pathogens and to avoid the risks associated with the emergence of resistance to new drugs.

## 1. Introduction

Antibiotic resistance is one of the main threats to human health, and for this reason, the World Health Organization (WHO) has recognized it as a global priority that needs to be addressed [1]. During the last two decades, there has been an increased prevalence of multidrug-resistant (MDR) Gram-negative bacteria, such as third-generation cephalosporin- and carbapenem-resistant *Enterobacterales* (CRE), carbapenem-resistant (CR) *Pseudomonas aeruginosa,* and CR *Acinetobacter baumannii*, which has led to higher morbidity and mortality rates [2,3,4,5,6]. The diffusion of carbapenem-resistant *Enterobacterales,* and in particular, of enterobacteria such as *Klebsiella pneumoniae* producing carbapenemase (KPC), New Delhi metallo-β-lactamase (NDM), and oxacillinase(OXA)-48-like enzymes, is a major concern for public health due to the reduced available treatment options and high case-fatality rates [7,8,9].

Italy has the highest number of infections and deaths due to multidrug-resistant bacteria in Europe [10]. There has been an alarming increase in resistance to carbapenems in several bacterial species in Italy, including *Klebsiella pneumoniae* (29.5% of isolates), *Pseudomonas aeruginosa* (15.9% of isolates), and *Acinetobacter baumannii* (80.8% of isolates). Among these species, *Klebsiella pneumoniae* producing the KPC enzyme is the most widespread bacterial strain resistant to antimicrobials in Italy. Since 2019, the appearance of isolates carrying NDM genes have been documented [11].

To overcome the issue of emerging antibiotic resistance, new drugs are needed. Cefiderocol, the first injectable siderophore cephalosporin approved by the FDA, was developed for this purpose [12]. Cefiderocol is a cephalosporin with high affinity for PBP 3 and its structure is similar to cefepime. Like other β-Lactams, it binds to bacterial penicillin binding proteins (PBP), inhibiting peptidoglycan synthesis. However, it is different from other β-Lactams in that it enters the bacterial periplasmic space through its siderophore-like activity. This drug presents an intrinsic structural stability not only to Ambler class A (e.g., KPC), C (e.g., AmpC), and D (e.g., OXA-48), but also to Amber class B β-lactamases (e.g., NDM, VIM, and IMP). In fact, cefiderocol is the first active antimicrobial against metallo-β-lactamases. Its molecular structure derives from ceftazidime and cefepime, with a carboxypropanoxyimino group in the side chain at position 7 of the cephem nucleus, improving stability to β-lactamases similar to ceftazidime, and a pyrrolidinium group at position 3 of the side chain, avoiding recognition by β-lactamases similar to cefepime. Additionally, cefiderocol has a catechol group in position 3 of the side chain, which is responsible for the siderophore-like property. This catechol group is recognized by transporters such as CirA and Fiu in *E. coli* or PiuA in *P. aeruginosa*, allowing cefiderocol to accumulate in the periplasmic space (“Trojan horse”) and avoid resistance mechanisms such as loss of porins. Altogether, these structural features enable this new antibiotic to be active against MDR Gram-negative bacteria such as *Pseudomonas aeruginosa*, *Acinetobacter baumannii*, *Stenotrophomonas maltophilia,* and *Enterobacterale*s that producing β-lactamases, including AmpC and extended-spectrum β-lactamases, or carbapenemases. In particular, cefiderocol activity against KPC-producing *Klebsiella pneumoniae* is comparable or higher than that of ceftazidime/avibactam [13]. Overall, cefiderocol appears to cause minor adverse effects such as diarrhea (4%), infusion site pain (3%), headache (3%), nausea (2%) [12].

In light of the emergence of bacteria resistant to different antibiotic classes, the aim of the current study was to evaluate in vitro susceptibility of different multi-drug resistant Gram-negative bacteria isolates responsible for different infections to cefiderocol. Although cefiderocol seems promising for treating severe infections caused by MDR bacteria such as complicated urinary infections and lower respiratory infections, some cases of resistance to this molecule due to mutations in iron transporter genes are documented in the literature [14,15]. Therefore, in order to analyze the possible mechanism of cefiderocol resistance, we performed whole-genome sequencing (WGS) and relative genomic analysis of two cefiderocol-resistant *Klebsiella pneumoniae* isolates.

## 2. Results

### 2.1. Antimicrobial Activity of Cefiderocol against MDRGram-Negative Bacteria

For all the clinical isolates included in this study, MIC distributions of cefiderocol are shown in Figure 1. According to the EUCAST criteria, 94% of the isolates were susceptible, while six percent of them were resistant.

Resistant isolates consisted of six *Klebsiella pneumoniae* and one *Escherichia coli* strains.

Out of the 56 *Klebsiella pneumoniae* strains, 37 isolates were KPC producers, three isolates were OXA-48 producers, one isolate produced NDM, 14 isolates co-produced two carbapenemases (OXA-48 and NDM), and one exhibited a membrane impermeability phenotype. Out of the eight *Escherichia coli* strains, four isolates were NDM producers, three isolates were AmpC producers, and one isolate was ESBL positive. Out of the three *Enterobacter* spp. strains, two isolates were VIM producers and two isolates coproduced OXA-48 and KPC carbapenemases. Concerning non-fermenting Gram-negative bacteria, out of the 33 *Pseudomonas aeruginosa* strains, four isolates were IMP producers, two isolates showed membrane impermeability, and one isolate produced VIM. For 26 isolates, the mechanism of resistance was not identified by the implemented routine assays. For *Acinetobacter baumannii*, *Achromobacter xylosoxidans,* and *Stenotrophomonas maltophilia*, the antimicrobial resistance mechanisms were not evaluated.

All of the isolates displayed a multidrug antimicrobial resistance phenotype because they were resistant to different antibiotic classes (cefepime, ceftriaxone, ceftazidime, aztreonam, amikacin, ciprofloxacin, trimethoprim/sulfamethoxazole, imipenem, meropenem, ceftazidime/avibactam, and ceftolozane/tazobactam). MIC50 and MIC90 values of all tested antibiotics against the bacterial isolates are shown in the Table 1.

Cefiderocol MICs ranged from 0.12 to 32 μg/L for Enterobacterales (from 0.03 to 2 μg/mL for *Klebsiella pneumoniae*, from 0.03 to 32 μg/mL for *Escherichia coli*, and from 0.12 to 2 μg/mL for *Enterobacter* spp.). Against *Klebsiella pneumoniae* isolates, the cefiderocol concentration able to inhibit 90 percent of the tested isolates was 2 μg/mL, and the susceptibility rate to cefiderocol was higher than those of the other antibiotics such as ceftozolane-tazobactam, ceftazidime, and ceftazidime-avibactam, but was lower than colistin. For non-fermenting Gram-negative bacteria, cefiderocol MICs ranged from 0.03 to 2 μg/mL (from 0.03 to 3 μg/mL for *Pseudomonas aeruginosa*, <2 μg/mL for *Acinetobacter baumannii*, 0.06 μg/mL for *Achromobacter xilosoxidans*, and from 0.5 to <2 μg/mL for *Stenotrophomonas maltophilia*). All non-fermenting Gram-negative bacteria were fully susceptible to cefiderocol.

### 2.2. Genomic Characteristics of Two Cefiderocol Resistant Klebsiella pneumoniae Isolates

To analyze the molecular mechanism underlining cefiderocol resistance, we performed a whole-genome sequencing analysis of two *Klebsiella pneumoniae* isolates (BSKP542 and BSKP713) responsible for invasive infections with a cefiderocol MIC of 4 μg/mL. The isolates were resistant to all of the tested antibiotics except for colistin and the aztreonam-avibactam combination.

The full length of BSKP542 and BSKP713 was 5,722,475 bp and 5,727,485 bp, respectively. Both isolates had 56.67% CG and 83 contigs of size ≥ 5000 bp. Both harbored the polysaccharide capsule KL30 and lipopolysaccharide O antigen O1V2 loci. BSKP542 and BSKP 713 were both assigned to the sequence type 383 (*gapA-infB-mdh-pgi-phoE-rpoB-tonB* allele number 2-6-1-3-8-1-18).

The genome of both isolates consisted of a chromosomal backbone, carrying all the virulence factors and some resistance genes, and five plasmids (Table 2a,b). In particular, 46 putative virulence factors and two efflux pump genes were detected in both genomes, while 35 and 33 resistance genes were identified in the BSKP542 and BSKP713 genomes, respectively. A virulence score of 0 was assigned to both our isolates because neither carried yersiniabactin (*ybt*), aerobactin (*iuc*), or colibactin (*clb*) [16]. The plasmids identified with an identity >91.6% were: pHAD28 (KU674895), pNDM-MAR (JN420336.1), pKpQIL-IT (JN233705), pKPN3 (CP000648), pOXA-48 (JN626286), and pNDM-5-IT (MG649062), which were previously described by Tian et al. in 2020 [17].

Due to the short-read output, we were unable to retrieve the complete antibiotic resistance plasmid sequences from both genomes. In particular, *bla*_NDM-1_, *bla*_CTX-M-15_, *qnrB1*, *aac(6′)-1b’,* and *cat* genes, which are respectively responsible for carbapenem, β-lactam, quinolone, aminoglycoside, and chloramphenicol resistance, were probably located on the pNDM-MAR plasmid from *Klebsiella pneumoniae*, showing 99.5% of identity and 100% of identical sites with BSKP542 and BSKP713 genomes (Table 2a). The pHAD28 plasmid from *Salmonella enterica*, subspecies *enterica serovar Hadar* strain HAD28 shared a minor identity percentage (91.6%) with our isolate genomes and probably carried the *qnrB19* gene coding for quinolones resistance. The pKpQIL-IT plasmid from the *Klebsiella pneumoniae* strain ST258 shared the highest identity and percentage of identical sites (both 100%) with the BSKP542 and BSKP713 genomes. Moreover, the pKpQIL-IT plasmid probably harbored the *bla*_TEM-1_ gene accounting for β-lactams resistance. Additionally, the *bla*_OXA-48_ gene was presumably localized on the pOXA-48 plasmid, characterized by 100% identity and 74.3% identical sites with the *Klebsiella pneumoniae* isolates of study. Although the pKPN3 plasmid was present in both isolates, it did not carry any antimicrobial resistance genes. The pNDM-5-IT plasmid, first described in *Escherichia coli strain Ec001,* shared 90.4% and 90.6% of identical sites withBSKP713 and BSKP542, respectively. The pNDM-5-IT plasmid also harbored many genes responsible for resistance to tetracycline, aminoglycosides, carbapenems, suphonamides, and macrolides, including *tet(A), aadA2*, *bla*_NDM-5_, *sul1*, *mphE,* and *mphA* (Table 2a).

### 2.3. Analysis of Mutations and Alterations in Genes Involved in Iron Uptake and Transport Detected in Two Cefiderocol-Resistance Klebsiella pneumoniae Isolates

In order to identify possible mutations which could contribute to cefiderocol resistance, the genes involved in iron uptake and transport were compared to reference genomes. All of the analyzed genes are reported in the Table 3.

Mutations in these genes are associated with cefiderocol resistance in the literature [18,19,20]. Several missense mutations were observed in a number of genes involved in iron uptake, such as *fhu*A, *fep*A, and *iut*A. All of the mutations identified were the same in both isolates. Furthermore, the same missense mutations in both isolates were detected for some genes involved in iron transport, such as *cir*A, *sit*C, *apb*C, *fep*G, *fep*C, *fet*B, *yic*I, *yic*J, and *yic*L. Two mutations, (L366H) and (R289C), were found in the *bae*S and *env*Z genes, which are responsible for the production of sensor kinases for two component systems able to regulate the expression of *cir*A and *fiu* iron uptake genes [19,21]. Mutations in these two genes have been previously associated with cefiderocol resistance [19].

Interestingly, in addition to other missense mutations, we found a premature stop codon (G569A) in the *fec*A gene leading to a truncated protein (Figure 2) and an insertion of four amino acids (PKPK) after lysine in the tonB protein (Table 3). FecA transports ferric iron across the outer membrane, and tonB confers the ability to acquire iron through all TonB-dependent iron uptake systems.

### 2.4. Phylogenetic Analysis

To deepen our understanding of the evolutionary relationships of BSKP542 and BSKP713 isolates among all the other *Klebsiella pneumoniae* isolates belonging to ST383 available in the Pathogen Watch database, we implemented a parsimony phylogenetic tree (Figure 3). Interestingly, our isolates clustered together, and they shared a common ancestor with a cluster formed by one isolate from Canada, two from Lebanon, and six from the Netherlands. Of note, 80.7% of isolates belonging to ST 383 originated in Europe, 14.2% were from Asia, 3.8% from North America, and 3.8% from Australia.

## 3. Discussion

This study shows that cefiderocol has an antimicrobial activity superior to that of comparators, including the new beta-lactam/beta-lactamase inhibitor combination drugs, such as ceftazidime-avibactam or ceftolozane-tazobactam, against isolates of ESBL and/or carbapenemase-producing *Enterobacterales*, carbapenem-non susceptible *Pseudomonas aeruginosa*, MDR *Acinetobacter baumannii,* and MDR *Stenotrophomonas maltophilia*. The cefiderocol susceptibility rates based on the EUCAST breakpoints for *Klebsiella pneumoniae*, *Escherichia coli*, *Enterobacter* spp, *Pseudomonas aeruginosa*, *Acinetobacter baumannii*, *Stenotrophomonas maltophilia,* and *Achromobacter xylosoxidans* were 88%, 89%, 100%, 100%, 100%, 100%, and 100%, respectively. A concentration of 2 μg/mL inhibited 94% of the isolates.

The activity of cefiderocol was lower among isolates belonging to the *Enterobacterales* group, and all of the resistant isolates were included. The overall observed resistance rate was 6.3%, but among the *Enterobacterales*, it reached 10.4%, a higher percentage compared that reported by other studies [22,23,24,25]. However, several studies have shown that the resistance rate of metallo-β-lactamases (MBL)-producing *Enterobacterales* is higher compared to non-MBL producers [26,27,28]. The resistance rate reported for NDM-producing pathogens ranged from 48.6% to 59%, with intervals of MIC90 values from 32 μg/mL to >64 μg/mL [26,27,29]. In addition to NDM-producers, VIM-producer strains also showed high resistance rates [26,27].

The carbapenemases produced by our resistant isolates were KPC (4 isolates) with MICs ranging from 4 μg/mL to >32 μg/mL, NDM (one isolate) with a MIC of 32 mg/mL, and NDM+OXA-48 (two isolates) with a MIC of 4 μg/mL. Resistance rates of 8.9% and 16.4%, based on the EUCAST breakpoint, were found in two studies from Europe in KPC-producing pathogens [26,27].

For *Enterobacterales* that produced β-lactamase OXA-48, a resistance rate of 7.1% and 11.8% was reported, while AmpC-positive pathogens were found to be susceptible to cefiderocol; this last finding was further confirmed by our study [26,27].

Our data show that cefiderocol had excellent activity against *Pseudomonas aeruginosa* with a MIC90 value of 2 μg/mL, consistent with the results of other studies [30,31,32,33,34]. We did not find a correlation between the presence of a particular carbapenemase and an increase in MICs against cefiderocol, as found in other studies [26,27].

Cefiderocol has a good activity also against *Stenotrophomonas maltophilia,* with MIC values ranging from 0.5 to <2 μg/mL. This range was higher than the one found in other studies, which ranged from 0.12 to 0.5 μg/mL [22,23,34]. However, due to the multiple resistance mechanisms present in this microorganism such as aminoglycoside-modifying enzymes, the L1 metallo-β-lactamase, the L2 serine active site β-lactamase [35], and the limited choice of effective drugs [36], cefiderocol seems to be a promising therapeutic option.

Even though only a few isolates were included in this study, *Achromobacter xylosoxidans* and *Acinetobacter baumannii* were susceptible to cefiderocol with MIC values of 0.06 μg/mL and <2 μg/mL, respectively.

Although the short-term usage of cefiderocol has been reported to cause strains with reduced susceptibility or resistance, primary resistance is infrequent. It is recommended to test susceptibility before the beginning of therapy, and to detect MIC changes, it is strongly recommended to test for strains isolated from patients treated with cefiderocol

Different resistant mechanisms have been proposed. For example, amino acid changes in the R2 loop of AmpC β-lactamases have been associated with a decreased susceptibility to cefiderocol [37,38]. Mutations in genes encoding TonB-dependent receptors, which are important for cefiderocol import, such as *PirA, PiuA*, and *PiuD*, have been detected in *Acinetobacter baumannii* and *Pseudomonas aeruginosa* [39,40]. The production of NDM and PER lactamases has been reported to contribute to an increase in cefiderocol MICs, but not to cefiderocol resistance [41]. Furthermore, mutations in the *cirA* gene, encoding the catecholate receptor, have been detected in resistant NDM-producing *Enterobacter cloacae* [42].

Because our strains showed cefiderocol resistance regardless of the beta lactamases harbored, we explored different iron transporter genes to identify mutations that could be associated with cefiderocol resistance. Our results revealed the presence of various mutations located in different genes involved in iron uptake and transport, such as *fhu*A, *fep*A, *iut*A, *cir*A, *sit*C, *apb*C, *fep*G, *fep*C, *fet*B, *yic*I, *yic*J, and *yic*L. These mutations were the same in both sequenced strains. Furthermore, we identified, for the first time to the best of our knowledge, a nonsense mutation in the *fecA* gene that gives rise to a premature stop codon and a truncated protein. This protein is a ferric iron citrate receptor, which represents another pathway by which bacteria may acquire iron. Its function is to transport ferric iron across the outer membrane, and it has been reported that the fecA system contributes to the virulence of enterobactin-mutant *Escherichia coli* [43]. Moreover, we found a 4-aminoacid insertion in the tonB protein. Generally, the TonB-dependent receptors are located in the outer membrane of Gram-negative bacteria and are important for the entry of cefiderocol into the periplasmic space.

Future studies are needed to confirm and elucidate the role of cefiderocol resistance in all these newly described mutations, and in particular, those mutations found in the *fecA* and *TonB* genes. Both isolates belonged to ST-383, a sequence type predominantly found in Europe.

This study presents many limitations. First, it included isolates from a single center and from only one geographic region. Moreover, our collection was small, and some Gram-negative bacteria were represented by only a few isolates. Larger studies with a greater number of isolates from more centers and different geographical areas are necessary to generalize these results.

## 4. Materials and Methods

### 4.1. Isolates

A total of 110 clinical isolates were included in this study. Only one isolate per patient infection episode was included. The strains were isolated from different clinical specimens collected between 2021 and 2022 in the Laboratory of Microbiology, Spedali Civili, Brescia, Italy. The samples consisted of needle aspirate, *n* = 3; bronchoalveolar lavage, *n* = 13; broncho aspirate, *n* = 7; biopsy specimens, *n* = 5; blood cultures, *n* = 26; sputum samples, *n* = 5; peritoneal fluid, *n* = 1, intra-abdominal fluid, *n* = 4; bile fluid, *n* = 1; pus, *n* = 1; skin swabs, *n* = 6; tracheal aspirate, *n* = 19; wounds, *n* = 8; and urine, *n* = 11.

The clinical isolates included 67 *Enterobacterales*, 2 *Acinetobacter baumannii*, 1 *Achromobacter xylosoxidans*, 33 *Pseudomonas aeruginosa* and 7 *Stenotrophomonas maltophilia.* Of the 67 Enterobacterales isolated, 56 were *Klebsiella pneumoniae*, 8 were *Escherichia coli,* and 3 were *Enterobacter* species. The isolates were identified by MALDI-TOF (Matrix Assisted Laser Desorption/Ionization; BioMérieux, Florence, Italy).

### 4.2. Antimicrobial Susceptibility Testing

Antimicrobial susceptibility tests were performed using an automatic microdilution broth (VITEK^®^2 system, BioMérieux). The Etest method (BioMérieux) was implemented for imipenem, meropenem, ceftazidime/avibactam, and ceftolozane/tazobactam susceptibility analysis, whereas the broth microdilution method was performed for colistin susceptibility testing, as recently recommended by the European Committee on Antimicrobial Susceptibility Testing (EUCAST) [44] and the Clinical and Laboratory Standards Institute (CLSI) [45].

Cefiderocol susceptibility was evaluated using an iron-depleted cation-adjusted Mueller-Hinton broth and plate (ID-CAMHB), with the both microdilution method performed using Sensititre^TM^ Cefiderocol MIC panel CMP1SHIH (Thermo Fisher Scientific, Monza, Italy) according to the manufacturer’s instructions. Briefly, 0.4 mL of a 1:20 diluted 0.5 McFarland bacterial solution was added to the ID-CAMHB tube provided in the kit. Subsequently, 100 µL of the obtained solution was dispensed into each well of the panel (range of cefiderocol concentration from 0.03 to 32 μg/mL). The panels were incubated at 35 °C for 20 h in ambient air before reading the MIC endpoints. ID-CAMHB did not significantly affect the growth of any quality controls or test organisms. The MIC of cefiderocol was read depending on the presence of strong growth in the ID-CAMHB growth control (i.e., a button of approximately 2 mm or greater). The cefiderocol MIC was read as the first panel of the well in which the isolate growth was significantly reduced (i.e., a button of <1 mm or light/faint turbidity) relative to the growth observed in the ID-CAMHB growth control well. The results were interpreted according to the latest EUCAST breakpoint (www.eucast.org/clinical_breakpoints, accessed on 2 January 2023) guidelines. The EUCAST PK/PD (non-species-related) susceptibility breakpoint of cefiderocol was applied to *Acinetobacter* spp. and *S. maltophilia* (MIC ≤ 2 μg/mL) because their species-specific clinical breakpoints were not available in EUCAST [46].

### 4.3. Phenotypic Detection of Carbapenemases

The O.K.N.V.I. Resist-5 immunochromatographic assay (Coris BioConcept, Gembloux, Belgium) was performed to efficiently detect class A (KPC), class B (VIM, IMP and NDM), and class D (OXA-48) carbapenemases according to the manufacturer’s instructions [47,48].

### 4.4. Whole Genome Sequencing (WGS)

Two *Klebsiella pneumoniae* isolates (BSKP542 and BSKP713) underwent whole-genome sequencing (WGS) to identify their genomic characteristics. Bacterial DNA was extracted from the bacterial colonies using a QIAmp DNA Mini Kit ^®^ (Qiagen, Hilden, Germany) according to the manufacturer’s instructions. The extracted bacterial DNA was quantified using the Qubit DNA HS Assay Kit (Thermo Fisher Scientific).

Whole genome libraries of the 2 bacterial isolates were prepared using the Nextera DNA Flex kit (Illumina, San Diego, CA, USA), and deep sequencing was performed on an Illumina MiSeq platform. Reads were checked for bacterial species confirmation using Kraken2 software [49]. Reads were trimmed with Trimmomatic version 0.39 [50] and assembled using SPAdes version 3.13.0 [51].

The online tool Pathogenwatch (https://pathogen.watch, accessed on 21 December 2021) was implemented to determine the in silico multi locus sequence typing and to analyze assemblies that passed quality control, with more than 95% of *Klebsiella* core genes detected. Sequence type definition was confirmed using the Bacterial Isolates Genome Sequence Database of the Pasteur Institute (https://bigsdb.pasteur.fr/klebsiella/, accessed on 19 April 2022). Antimicrobial resistance genes, virulence genes (including the derived virulence score), capsule, and lipopolysaccharide-type genes were identified using Kleborate via Pathogenwatch. To confirm the Pathogenwatch data, ResFinder 3.2 was used to detect antimicrobial resistance genes (https://cge.cbs.dtu.dk/services/ResFinder/, accessed on 24 May 2022). The presence of plasmids was detected using PlasmidFinder 2.0 (https://cge.cbs.dtu.dk/services/PlasmidFinder-2.0/, accessed on 9 November 2022). Virulent factors were determined by blasting the assembled the isolate genome against the Virulent Factor Database (VFDB) using the VFanalyser tool (http://www.mgc.ac.cn/VFs/, accessed on 7 January 2022). The presence of mobile genomic elements in plasmids was assessed using the Mobile Element Finder (https://cge.cbs.dtu.dk/services/MobileElementFinder/, accessed on 9 November 2022), then confirmed manually by mapping the raw reads using Bowtie2 implemented in Geneious software (v. 11.1.5) (Biomatters Ltd., Auckland, New Zealand). The kSNP3 program was implemented to build a parsimony phylogenetic tree [52] of BSKP542 and BSKP713 isolates, along with all the other isolates of ST-383 available in Pathogenwatch.

## 5. Conclusions

Our data show that cefiderocol is an effective drug against most of the analyzed multidrug-resistant Gram-negative bacteria, with a susceptibility rate of 94%. However, the presence of a higher resistance rate in carbapenem-resistant *Enterobacterales* highlights the importance of active surveillance, including genomic surveillance through rapid molecular testing. Such testing can quickly identify and differentiate between carbapenemases genes, allowing for appropriate therapy and avoiding the spread of these difficult-to-treat microorganisms. Maintaining the activity of novel drugs will consequently reduce the risks associated with emerging drug resistance.

## Figures and Tables

**Figure 1 antibiotics-12-00785-f001:**
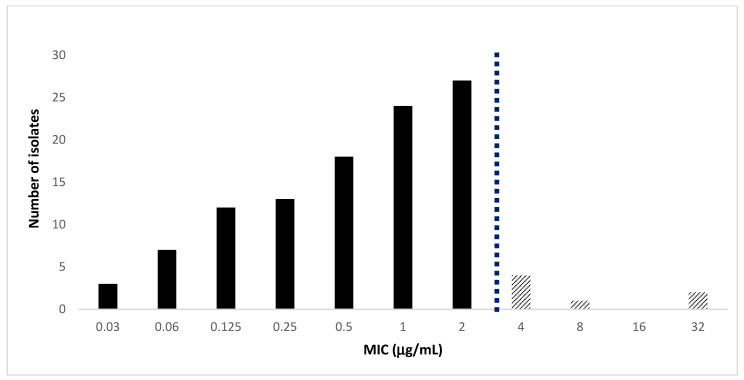
Distribution of cefiderocol MICs for all of the isolates according to EUCAST breakpoints. Black column, susceptible isolates; striped columns, resistant isolates. The dashed bar separates susceptible and resistant isolates.

**Figure 2 antibiotics-12-00785-f002:**
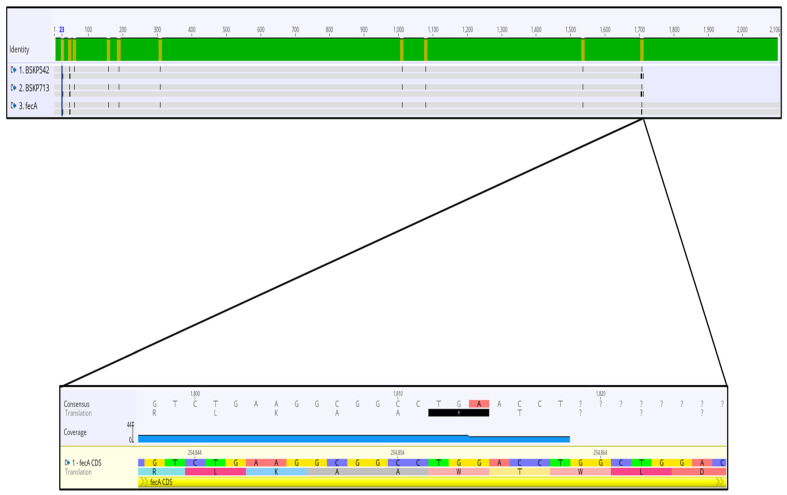
Presence of nonsense mutations in the *fecA* gene in BSKP713 and BSKP542 isolates. A nucleotide mutation from G to A in position 254,858 generated a TGA stop codon in the *fecA* gene in both of our isolates.

**Figure 3 antibiotics-12-00785-f003:**
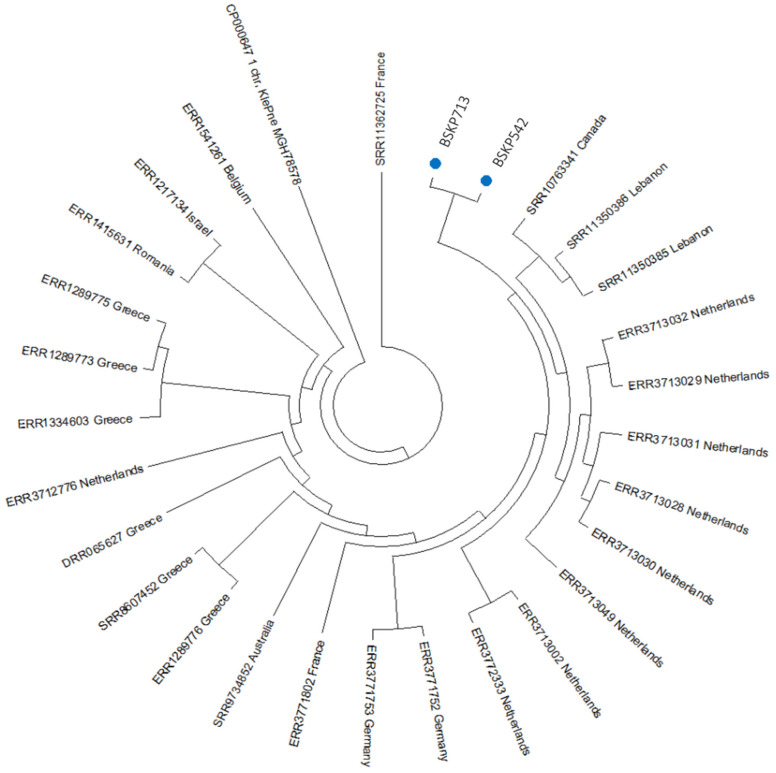
Phylogenetic tree showing evolutionary relationships among BSKP542 and BSKP713 isolated in this study and all the other isolates that also belonged to ST383. Our isolates are shown with blue circles. *Klebsiella pneumoniae* ATCC 700721 was employed as an outgroup.

**Table 1 antibiotics-12-00785-t001:** In vitro activity of antibiotics against Gram-negative bacteria collected during the study period.

Organisms (N° of Isolates)	Drug	MIC Range (μg/mL)	MIC_50_	MIC_90_	Susceptible ^a^ (%)	Resistant ^a^ (%)
*K. pneumoniae* (*n* = 56)	CFDC	0.03–4	1	2	89	11
	AN	<1–32	32	32	27	73
	AMC	>16	>16	>16	0	100
	FEP	0.12 to >16	>16	32	0	100
	CT	>8 to >16	>16	>16	0	100
	CAZ	>32	>32	>32	0	100
	CZA	0.5 to >16	>16	>16	29	71
	CIP	<0.06 to >2	>2	>2	4	96
	GEM	<1 to >8	2	>8	52	48
	IPM	<0.25 to >8	>8	>8	2	98
	MEM	4 to >128	>8	>128	2	98
	TZP	>64	>64	>64	0	100
	CST	<0.5 to >8	0.5	1	98	2
	SXT	<20 to >160	>160	>160	20	80
*E. coli* (*n* = 8)	CFDC	0.03–32	NC	NC	88	12
	AN	2–16	NC	NC	88	12
	AMC	>16	NC	NC	0	100
	FEP	0.12 to >16	NC	NC	12	88
	CT	<0.25 to >16	NC	NC	50	50
	CAZ	8 to >32	NC	NC	0	100
	CZA	0.12 to >8	NC	NC	50	50
	CIP	>2	NC	NC	0	100
	GEM	<1 to >8	NC	NC	63	37
	IPM	<0.25 to >8	NC	NC	50	50
	MEM	<0.25 to >8	NC	NC	50	50
	TZP	<4 to >64	NC	NC	37	63
	CST	0.5	NC	NC	100	0
	SXT	<20 to >160	NC	NC	12	88
*Enterobacter* spp. (*n* = 3)	CFDC	0.12–2	NC	NC	100	0
	AN	<1	NC	NC	100	0
	AMC	>16	NC	NC	0	100
	FEP	>16	NC	NC	0	100
	CT	>16	NC	NC	0	100
	CAZ	>32	NC	NC	0	100
	CZA	>8	NC	NC	0	100
	CIP	1 to >2	NC	NC	0	100
	IPM	>8	NC	NC	0	100
	MEM	>8	NC	NC	0	100
	TZP	>64	NC	NC	0	100
	CST	0.5	NC	NC	100	0
	SXT	>160	NC	NC	0	100
*P. aeruginosa* (*n* = 33)	CFDC	0.03–2	1	2	100	0
	AN	<1 to >32	4	>32	84	16
	FEP	4 to >16	>16	>16	0	100
	CT	0.5 to >64	>8	>16	40	60
	CAZ	2 to >32	>32	>32	0	100
	CZA	2 to >64	>8	>8	28	72
	CIP	0.5 to >2	>2	>2	0	100
	IPM	2 to >8	>8	>8	0	100
	MEM	1 to >8	>8	>8	0	100
	TZP	32 to >64	>64	>64	0	100
	CST	<0.5 to >8	0.5	2	94	6
*A. baumannii* (*n* = 2)	CFDC	<=2	NC	NC	100	0
	AN	>32	NC	NC	0	100
	FEP	>16	NC	NC	0	100
	TIG	1	NC	NC	NA	NA
	CAZ	>32	NC	NC	0	100
	ATM	>256	NC	NC	NA	NA
	CIP	>2	NC	NC	0	100
	IPM	>8	NC	NC	0	100
	MEM	>8 to >256	NC	NC	0	100
	TZP	>64	NC	NC	0	100
	CST	0.5	NC	NC	100	0
*A. xylosoxidans* (*n* = 1)	CFDC	0.06	NC	NC	100	0
	AN	>16	NC	NC	NA	NA
	AMC	>16	NC	NC	NA	NA
	FEP	>16	NC	NC	NA	NA
	CT	>64	NC	NC	NA	NA
	CAZ	16	NC	NC	NA	NA
	CZA	16	NC	NC	NA	NA
	CIP	>1	NC	NC	NA	NA
	GEM	>8	NC	NC	NA	NA
	MEM	8	NC	NC	0	100
	TZP	>128	NC	NC	0	100
	CST	2	NC	NC	NA	NA
	SXT	<1	NC	NC	NA	NA
*S. maltophilia* (*n* = 7)	CFDC	0.5 to <2	NC	NC	100	0
	AN	>32	NC	NC	NA	NA
	CT	>32	NC	NC	NA	NA
	CAZ	>32	NC	NC	0	100
	CZA	>16	NC	NC	NA	NA
	MEM	>32 to >64	NC	NC	NA	NA
	CST	>8	NC	NC	NA	NA
	SXT	0.01 to >4	NC	NC	71	29

Drug abbreviations: CFDC, cefiderocol; AN, amikacin; AMC, amoxicillin-clavulanic acid; FEP, cefepime; CT, ceftolozane-tazobactam; CAZ, ceftazidime; CZA, ceftazidime-avibactam; CIP, ciprofloxacin; GEM, gentamicin; IPM, imipenem; MEM, meropenem; TZP, piperacillin-tazobactam; CST, colistin; SXT, trimethoprim-sulfamethoxazole; ATM, aztreonam. NA, not applicable; NC, not calculated because the number of isolates was <10. ^a^ EUCAST breakpoints.

**Table 2 antibiotics-12-00785-t002:** (a) Antimicrobial resistance (AMR) determinants of the two cefiderocol-resistant *Klebsiella pneumoniae* isolates. (b) Virulence determinants of the two cefiderocol-resistant *Klebsiella pneumoniae* isolates.

(a)					
Isolates	ST	Serotype	AMR Determinants		
			Gene Function	Detected Genes	Hypothetical Location
BSKP542	383	HL30, O1V2	aminoglycoside resistance	*aac (6′)-1b’*	pNDM-MAR
				*aadA2*	pNDM-5-IT
				*aph(3′)-VI*	chromosome
				*armA*	chromosome
			beta-lactam resistance	*bla* _NDM-1_	pNDM-MAR
				*bla* _CTX-M15_	pNDM-MAR
				*bla* _TEM-1b_	pKpQIL-IT
				*bla* _Tem-1c_	pKpQIL-IT
				*bla* _OXA-48_	pOXA-48
				*bla* _NDM-5_	pNDM-5-IT
				*bla* _SHV-26_	chromosome
				*bla* _SHV-78_	chromosome
				*bla* _SHV-98_	chromosome
				*bla* _SHV-179_	chromosome
				*bla* _SHV-145_	chromosome
				*bla* _SHV-194_	chromosome
				*bla* _SHV-199_	chromosome
				*bla* _OXA-9_	chromosome
				*bla* _CTX-M14b_	chromosome
				*ompK36*	chromosome
				*ompK37*	chromosome
			macrolide resistance	*mph(A)*	pNDM-5-IT
				*mph(E)*	pNDM-5-IT
				*msr(E)*	chromosome
			quinolone resistance	*qnrB19*	pHAD28
				*qnrB1*	pNDM-MAR
				*qnrS1*	chromosome
				*parC*	chromosome
				*gyrA*	chromosome
			tetracycline resistance	*tet(A)*	pNDM-5-IT
			fosfomycin resistance	*fosA*	chromosome
			phenicol resistance	*catA1*	pNDM-MAR
			trimethoprim resistance	*dfrA5*	chromosome
			sulphonamide resistance	*sul1*	pNDM-5-IT
				*sul2*	chromosome
BSKP713	6339	HL30, O1V2	aminoglycoside resistance	*aac (6′)-1b’*	pNDM-MAR
				*aadA2*	pNDM-5-IT
				*aph(3′)-VI*	chromosome
				*armA*	chromosome
			beta-lactam resistance	*bla* _NDM-1_	pNDM-MAR
				*bla* _CTX-M15_	pNDM-MAR
				*bla* _TEM-1b_	pKpQIL-IT
				*bla* _Tem-1c_	pKpQIL-IT
				*bla* _OXA-48_	pOXA-48
				*bla* _NDM-5_	pNDM-5-IT
				*bla* _SHV-26_	chromosome
				*bla* _SHV-78_	chromosome
				*bla* _SHV-98_	chromosome
				*bla* _SHV-179_	chromosome
				*bla* _SHV-145_	chromosome
				*bla* _SHV-194_	chromosome
				*bla* _SHV-199_	chromosome
				*bla* _OXA-9_	chromosome
				*bla* _CTX-M14b_	chromosome
				*ompK36*	chromosome
				*ompK37*	chromosome
			macrolide resistance	*mph(A)*	pNDM-5-IT
				*mph(E)*	pNDM-5-IT
				*msr(E)*	chromosome
			quinolone resistance	*qnrB19*	pHAD28
				*qnrB1*	pNDM-MAR
				*qnrS1*	chromosome
			tetracycline resistance	*tet(A)*	pNDM-5-IT
			fosfomycin resistance	*fosA*	chromosome
			phenicol resistance	*catA1*	pNDM-MAR
			trimethoprim resistance	*dfrA5*	chromosome
			sulphonamide resistance	*sul1*	pNDM-5-IT
				*sul2*	chromosome
**(b)**					
**Isolates**			**Virulence Determinants**		
			**Gene Function**	**Detected Genes**	
BSKP542 and BSKP713			**Adhesion**		
			type 1 fimbriae	*mrk*ABCDFHJ	
			type 3 fimbriae	*fim*ABCDEFGHIK	
			type 4 pili	*pil*W	
			fimbrial adherence determinants	*stb*ABCDE	
			**Iron uptake**		
			aerobactin	*iut*A	
			ent siderophore	*ent*ABCDEFS; *fep*ABCDG; *fes*	
			salmochelin	*iro*EN	
			**Regulatory system**	*rcs*AB	
			**Secretion system**	*T6SS (I-III)*	
			**Efflux pump genes**		
			RND efflux pump	*arc*AB	

**Table 3 antibiotics-12-00785-t003:** Analysis of possible mutations in genes involved in iron uptake and transport systems.

Genes	Function	Mutations/Alterations	
		**BSKP542**	**BSKP713**
*fhu*A	Iron uptake [18]	V176F, I178V, I212L, G269D, V609G	V176F, I178V, I212L, G269D, V609G
*fep*A	Iron uptake [18]	P531A	P531A
*fbp*A	Iron uptake [18]	WT	WT
*efe*O	Iron uptake [18]	WT	WT
*exb*B	Iron uptake [18]	WT	WT
*exb*D	TonB-dependent energy transduction system reported to affect the function of Iron transporters [18]	WT	WT
*fiu*A	Iron uptake [18]	Absent	Absent
*fur*	Iron uptake [18]	WT	WT
*iut*A	Iron uptake [18]	E160K, S285P	E160K, S285P
*bae*S	Encodes a sensor kinase protein of the two-component BaeSR signal transduction system [18,19]	L366H	L366H
*env*Z	Two-component transcriptional regulator reported to affect the expression of iron transporters [18,19]	R289C	R289C
*cir*A	Encodes receptor which preferentially transports catecholate siderophores [18,19]	A134V, N558D	A134V, N558D
*feo*A	Ferrous iron uptake [18]	WT	WT
*sit*C *	Iron/manganese ABC transporter permease subunit [18]	Y11H, R166C	Y11H, R166C
*apb*C	Iron-sulfur cluster carrier protein [18]	P334S, I339T	P334S, I339T
*fep*G	Iron-enterobactin ABC transporter permease [18]	V63M, G206S	V63M, G206S
*fep*C	Iron-enterobactin ABC transporter ATP-binding protein [18]	T9A, I182N	T9A, I182N
*fet*B	Iron export ABC transporter permease subunit FetB [18]	WT	WT
*fet*A	Iron ABC transporter ATP-binding protein FetA [18]	S27N	S27N
*fec*A	TonB energy transducing system-dependent ferric citrate uptake receptor [20]	A9V, V16L, premature STOP codon in position aa569	A9V, V16L, premature STOP codon in position aa569
*ton*B	Component of inner membrane protein complex providing energy to TonB dependent transporters [19]	G60A, A80V, insertion of 4 aa in position 103 (PKPK), P168A, E219Q	G60A, A80V, insertion of 4 aa in position 103 (PKPK), P168A, E219Q
*fiu*	Encodes receptor that preferentially transports catecholate siderophores [18,19]	D387N	D387N
*omp*R	Two-component transcriptional regulator reported to affect the expression of iron transporters [19]	A72V	A72V
*yic*I	Transporter family [18,19]	S30N, L323Q, K358N, G624C, H653R	S30N, L323Q, K358N, G624C, H653R
*yic*J	Transporter family [18,19]	P102L, F181Y	P102L, F181Y
*yic*L	Transporter family [18,19]	WT	WT
*chr*A *	Heavy metals transporter [19,20]	A245V, G344A	A245V, G344A

Reference sequence, *Klebsiella pneumoniae* ATCC 13883; * reference sequence, *Klebsiella pneumoniae* CP071027.

## Data Availability

Data are available at EBI under study accession n. PRJEB60070.

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
