# Peer review of "In Vitro Activity of Cefiderocol on Multiresistant Bacterial Strains and Genomic Analysis of Two Cefiderocol Resistant Strains"

_antibiotics, 2023, doi:10.3390/antibiotics12040785_

Round 1

Author Response

Answers to reviewer 1:

 Line 276-284 : these results are interesting. As the author says, it would be necessary to study the behavior of cefiderocol on a larger number of Klebsiella, E. coli, Pseudomonas Acinetobacter baumanii, which come from other localities, and even wild strains.

We agree with the reviewer and we have no answer for this comment.

Line 287-288: You are talking about which resistance mechanisms and which type of resistance. Because there is acquired resistance and natural resistance. Some acquired resistance mechanisms are different from natural resistance mechanisms. You need to be more clear.

Another thing is how you determined these mechanisms. Before talking about the most common mechanisms in Gram-negative organisms, you should first clarify how you identified these mechanisms in clinical isolates.

To my knowledge, in Gram-negative bacteria, three mechanisms are the most used: Enzyme inhibition, efflux pumps and horizontal gene transfer. What are you talking about?

We agree with the reviewer’s observation. We just refer to acquired resistance and principally to the production of different carbapenemases and extended-spectrum beta-lactamases, which are the only mechanisms analyzed. Therefore, we have deleted this sentence from the paper.

Line 287-300: In general, when human bacterial isolates are used in a scientific study, their origin is specified. Most importantly, the study should have been approved by an ethics committee or the informed consents should have been signed by the owners of the isolates. Or a document from the administrative authorities authorizing this study should have been signed before the beginning of the study. I did not see these details in the materials and methods.

Ethical approval was not required for this study since the clinical samples were obtained within the routine microbiological diagnostics. In fact, in this study no patient’s specimens were analysed, only bacterial isolates from routine diagnostic culture. Precisely, samples were coded and anonymised and preserved as part of the routine diagnostics (standard care). Procedures performed in the study were in accordance with the ethical standards of the Institutional and National Research Committee and with the 1964 Helsinki Declaration and its later amendments or comparable ethical standards.

Patient consent was waived because clinical strains used in this study were retrospectively collected from the program of surveillance of healthcare-associated multi-drug resistant Gram-negative infections performed at our Microbiology Laboratory and no patient identification information was presented.

We have added this part in the dedicated Notes

Line 311-314: Please give a reference on which you relied ; Also it would be better to make a brief description of the method you used for the susceptibility test with Cefiderocol, because your description is not enough

According to the reviewer’s suggestion, a brief description of the method has been added.

 Line 321-324 : The reference you gave doesn't clearly explain the method you used to identify carbapenem resistance phenotypes. So either you give a little clear description or you give a reference that clearly explains this method

We have added the reference 48 that better explains the method PMID: 33921669

 Line 356-362: Although the results obtained are quite interesting, the increase in the rate of resistance to Cefiderocol in carbapenem-producing Enterobacteriaceae will occur if this antibiotic is widely used. So it's not just surveillance for the emergence of more resistance that should be done. Physicians should also be made aware of good antibiotic prescribing practices to avoid the emergence of resistance, such as prescribing two antibiotics with different modes of action at the same time, performing an antibiotic susceptibility test before prescribing an antibiotic.

We agree with the reviewer’s observation about the need to use combined antibiotics instead of the use of monotherapy to control the problem of antibiotic resistance, but it is also true, as shown in a recent paper that natural resistance against cefiderocol exists as it was found that there were more high-level cefiderocol-resistant isolates among carbapenem-resistant E. coli even before using it in therapy (PMID: 35481835) and this could be considered as an important alert for continuous surveillance.

Reviewer 2 Report

The study is well designed and provides valuable insights into the susceptibility of multidrug-resistant Gram-negative bacteria to cefiderocol. However, it would be helpful to provide more information on the clinical outcomes of patients infected with these organisms and treated with cefiderocol. This information could help to provide a more comprehensive understanding of the efficacy of cefiderocol in treating these infections.

 The authors mention the limitations of the study, including the small sample size and the fact that the isolates were from a single center and one geographic region. It would be interesting to see future studies that address these limitations and include more isolates from different regions to determine the generalizability of the findings.

 In the Methods section, it is mentioned that the immunochromatographic assay was used to detect carbapenemases. However, it is not clear how the results of this assay were used in the analysis. Providing more information on the role of carbapenemases in the susceptibility to cefiderocol would be helpful.

 In the discussion section, the authors mention the importance of active surveillance, including genomic surveillance, to avoid the spread of these difficult-to-treat microorganisms. However, it would be helpful to provide more information on the strategies that can be used for active surveillance and how these strategies can be implemented in clinical practice.

 Some minor and major comments and suggestion for the authors

Abstract:

 Line 2: "multi-drug resistant" should be "multidrug-resistant"

Line 4: "previously characterized pathogens" could be more specific about which pathogens were tested

Line 5: "microdilution panels" should be "microdilution assays"

Line 15: "MIC<2mg/mL" should be "MIC < 2 mg/mL"

Line 18: "cefiderocol resistant" should be "cefiderocol-resistant"

Line 23: "which" should be "that"

Line 24: "aminoacidic" should be "amino acid"

Introduction

Line 34: "world priority to address" could be rephrased as "a global priority for addressing"

Line 40: "New Delhi metallo- -lactamase" should be "New Delhi metallo-β-lactamase"

Line 41: "oxacillinase- (OXA)-48-like" should be "oxacillinase (OXA)-48-like"

Line 43-44: "Italy represents" could be rephrased as "Italy has the highest number of infections and death due to multidrug-resistant bacteria in Europe [10], with an alarming increase in resistance to carbapenems in several bacterial species, including Klebsiella pneumoniae (29.5% of isolates), Pseudomonas aeruginosa (15.9% of isolates), and Acinetobacter baumannii (80.8% of isolates)"

Line 47-49: "isolates carrying NDM genes have been documented" could be more specific about the prevalence of NDM-carrying isolates in Italy

Line 50-52: "injectable siderophore cephalosporin" could be more specific about the mechanism of action of cefiderocol

Line 57: "antimicrobic" should be "antimicrobial"

Line 66: "producing β-lattamases," Is the other means β-lactamases? Please correct

Line 60-61: "pyrrolidinium group in the side chain in 60" could be rephrased as "pyrrolidinium group in position 3 of the side chain"

Line 73: "in vitro-susceptibility" should be "in vitro susceptibility"

Line 74: "caused by MDR bacteria" could be more specific about the types of infections

Results:

Line 81: "MDR-Gram negative bacteria" should be "MDR Gram-negative bacteria."

Line 84: "6%" should be "six percent."

Line 85: "Klebsiella pneumoniae and 1 Escherichia coli strains" should be "Klebsiella pneumoniae and one Escherichia coli strain."

Line 87: "carbapenemases (OXA-48 +NDM)" should be "carbapenemases (OXA-48 and NDM)."

Line 88: "membrane impermeability" should be "membrane impermeability phenotype."

Line 93: "produced AmpC" should be "AmpC producers."

Line 94: "VIM producers and 2 isolates coproduced" should be "VIM producers, and two isolates co-produced."

Line 99: "Stenotrophomonas maltophilia the antimicrobial resistance mechanisms were not evaluated" Add comma after "Stenotrophomonas maltophilia,

Line 100: "antibiotic phenotype" should be "antimicrobial resistance phenotype."

Line 113: "2 g/mL" should be "two micrograms per milliliter." Please confirm it

Line 115: "90%" should be "90 percent."

Line 116-117: "it" should be "cefiderocol."

Line 121: "All of them were totally susceptible" could be rephrased as "All of them were fully susceptible."

In sentence 126, "with a MIC of 4 g/mL against cefiderocol" should be changed to "with a cefiderocol MIC of 4 μg/mL."

In sentence 129, "56,67% CG" should be changed to "56.67% GC."

In sentence 142, "already described by Tian et al., 2020" should be changed to "previously described by Tian et al. in 2020."

In sentence 155, "identical sites percentage" should be changed to "percentage of identical sites."

In sentence 156, "74,3% identical sites" should be changed to "74.3% identical sites."

In sentence 157, "it did not carry any antimicrobic resistance gene" should be changed to "it did not carry any antimicrobial resistance genes."

In sentence 159, "with, respectively, BSKP713 and BSKP542" should be changed to "with BSKP713 and BSKP542, respectively."

Sentence 172: "Different missense mutations were observed in a number of genes involved in iron uptake such as fhuA, fepA and iutA." - This sentence could be rewritten as "Several missense mutations were observed in genes involved in iron uptake, such as fhuA, fepA, and iutA.

 Discussion:

 Line 231-233: in the sentence "For Enterobacterales, which produced beta-lactamase OXA-48, a resistance rate of 7.1% and 11.8% was reported, while AmpC-positive pathogens were found all susceptible to cefiderocol; this finding was further confirmed in our study", it's not entirely clear what "this finding" refers to - does it refer to the susceptibility of AmpC-positive pathogens to cefiderocol, or to the resistance rate of Enterobacterales producing OXA-48? More context would help clarify this.

Line 276-277: In the sentence "Future studies are needed to confirm and elucidate the role played in cefiderocol resistance by these newly described mutations," it would be clearer to specify which mutations are being referred to.

Line 278-279: In the sentence "Both isolates, which underwent WGS belonged to ST-383, a sequence type principally found in Europe as highlighted by the phylogenetic analysis," it may be clearer to state that the isolates belong to a sequence type predominantly found in Europe.

 Methodology:

Section 4.1 Isolates and section 4.2 Antimicrobial susceptibility: The authors provide a clear and detailed description of the isolates and antimicrobial susceptibility testing methods used in the study, which enhances the reproducibility and transparency of their results.

Author Response

The study is well designed and provides valuable insights into the susceptibility of multidrug-resistant Gram-negative bacteria to cefiderocol. However, it would be helpful to provide more information on the clinical outcomes of patients infected with these organisms and treated with cefiderocol. This information could help to provide a more comprehensive understanding of the efficacy of cefiderocol in treating these infections.

We agree with the reviewer’s opinion about the utility of this information, but we cannot extrapolate these data because the study was performed on clinical strains retrospectively collected and from anonymized clinical samples. In this study our aim was only to evaluate the cefiderocol efficacy in vitro.

The authors mention the limitations of the study, including the small sample size and the fact that the isolates were from a single center and one geographic region. It would be interesting to see future studies that address these limitations and include more isolates from different regions to determine the generalizability of the findings.

We agree with the referee’s comment, and we hope to be able to further extend this baseline study.

In the Methods section, it is mentioned that the immunochromatographic assay was used to detect carbapenemases. However, it is not clear how the results of this assay were used in the analysis. Providing more information on the role of carbapenemases in the susceptibility to cefiderocol would be helpful.

All our strains were characterized for the presence of specific carbapenemases and we have reported the results obtained (lines 94-103) in the Results section, subsection 2.1. Since the presence of a particular carbapenemase was associated with different susceptibility to cefiderocol, we have discussed this in lines 227-244 of the Discussion section.

In the discussion section, the authors mention the importance of active surveillance, including genomic surveillance, to avoid the spread of these difficult-to-treat microorganisms. However, it would be helpful to provide more information on the strategies that can be used for active surveillance and how these strategies can be implemented in clinical practice.

We have added a brief sentence in the Conclusion section.

 Some minor and major comments and suggestion for the authors

Abstract:

 Line 2: "multi-drug resistant" should be "multidrug-resistant"

Corrected

Line 4: "previously characterized pathogens" could be more specific about which pathogens were tested

Corrected

Line 5: "microdilution panels" should be "microdilution assays"o

Corrected

Line 15: "MIC<2mg/mL" should be "MIC < 2 mg/mL"

Corrected

Line 18: "cefiderocol resistant" should be "cefiderocol-resistant"

Corrected

Line 23: "which" should be "that"

Corrected

Line 24: "aminoacidic" should be "amino acid"

Corrected

Introduction

Line 34: "world priority to address" could be rephrased as "a global priority for addressing"

Corrected

Line 40: "New Delhi metallo- -lactamase" should be "New Delhi metallo-β-lactamase"

Corrected

Line 41: "oxacillinase- (OXA)-48-like" should be "oxacillinase (OXA)-48-like"

Corrected

Line 43-44: "Italy represents" could be rephrased as "Italy has the highest number of infections and death due to multidrug-resistant bacteria in Europe [10], with an alarming increase in resistance to carbapenems in several bacterial species, including Klebsiella pneumoniae (29.5% of isolates), Pseudomonas aeruginosa (15.9% of isolates), and Acinetobacter baumannii (80.8% of isolates)"

Corrected

Line 47-49: "isolates carrying NDM genes have been documented" could be more specific about the prevalence of NDM-carrying isolates in Italy

Unfortunately, we cannot be more precise about the prevalence of NDM carrying isolates in Italy because few data on NDM-Kpn strains are available, also due to their sporadic spread. An outbreak of NDM-producing Enterobacterales occurred among patients without travel history hospitalized in the Tuscany region, Italy, with a total of 1645 cases of colonization or infection with NDM-CRE reported in the period from 1 November 2018 to 31 October 2019 and few cases were reported in 4 other regions (PMID: 32070467, PMID: 32070467, https://www.ecdc.europa.eu/sites/default/files/documents/04-Jun-2019-RRA-Carbapenems%2C%20Enterobacteriaceae-Italy.pdf. Accessed February4, 2021)

Line 50-52: "injectable siderophore cephalosporin" could be more specific about the mechanism of action of cefiderocol

We have added more sentences about the mechanism of action of cefiderocol

Line 57: "antimicrobic" should be "antimicrobial"

Corrected

Line 66: "producing β-lattamases," Is the other means β-lactamases? Please correct

Corrected

Line 60-61: "pyrrolidinium group in the side chain in 60" could be rephrased as "pyrrolidinium group in position 3 of the side chain"

Corrected

Line 73: "in vitro-susceptibility" should be "in vitro susceptibility"

Corrected

Line 74: "caused by MDR bacteria" could be more specific about the types of infections

We have specified the types of infections

Results:

Line 81: "MDR-Gram negative bacteria" should be "MDR Gram-negative bacteria."

Corrected

Line 84: "6%" should be "six percent."

Corrected

Line 85: "Klebsiella pneumoniae and 1 Escherichia coli strains" should be "Klebsiella pneumoniae and one Escherichia coli strain."

Corrected

Line 87: "carbapenemases (OXA-48 +NDM)" should be "carbapenemases (OXA-48 and NDM)."

Corrected

Line 88: "membrane impermeability" should be "membrane impermeability phenotype."

Corrected

Line 93: "produced AmpC" should be "AmpC producers."

Corrected

Line 94: "VIM producers and 2 isolates coproduced" should be "VIM producers, and two isolates co-produced." Corrected

Line 99: "Stenotrophomonas maltophilia the antimicrobial resistance mechanisms were not evaluated" Add comma after "Stenotrophomonas maltophilia,

Corrected

Line 100: "antibiotic phenotype" should be "antimicrobial resistance phenotype."

Corrected

Line 113: "2 g/mL" should be "two micrograms per milliliter." Please confirm it

Corrected

Line 115: "90%" should be "90 percent."

Corrected

Line 116-117: "it" should be "cefiderocol."

Corrected

Line 121: "All of them were totally susceptible" could be rephrased as "All of them were fully susceptible."

Corrected

In sentence 126, "with a MIC of 4 g/mL against cefiderocol" should be changed to "with a cefiderocol MIC of 4 μg/mL."

Corrected

In sentence 129, "56,67% CG" should be changed to "56.67% GC."

Corrected

In sentence 142, "already described by Tian et al., 2020" should be changed to "previously described by Tian et al. in 2020."

Corrected

In sentence 155, "identical sites percentage" should be changed to "percentage of identical sites."

Corrected

In sentence 156, "74,3% identical sites" should be changed to "74.3% identical sites."

Corrected

In sentence 157, "it did not carry any antimicrobic resistance gene" should be changed to "it did not carry any antimicrobial resistance genes."

Corrected

In sentence 159, "with, respectively, BSKP713 and BSKP542" should be changed to "with BSKP713 and BSKP542, respectively."

Corrected

Sentence 172: "Different missense mutations were observed in a number of genes involved in iron uptake such as fhuA, fepA and iutA." - This sentence could be rewritten as "Several missense mutations were observed in genes involved in iron uptake, such as fhuA, fepA, and iutA.

Corrected

 Discussion:

Line 231-233: in the sentence "For Enterobacterales, which produced beta-lactamase OXA-48, a resistance rate of 7.1% and 11.8% was reported, while AmpC-positive pathogens were found all susceptible to cefiderocol; this finding was further confirmed in our study", it's not entirely clear what "this finding" refers to - does it refer to the susceptibility of AmpC-positive pathogens to cefiderocol, or to the resistance rate of Enterobacterales producing OXA-48? More context would help clarify this.

We refer to the susceptibility of AmpC-positive pathogens to cefiderocol; we have corrected the sentence

Line 276-277: In the sentence "Future studies are needed to confirm and elucidate the role played in cefiderocol resistance by these newly described mutations," it would be clearer to specify which mutations are being referred to.

We refer to all the mutations reported in this study but in particular to mutations in fecA and TonB genes. We have added this sentence in the Discussion section

Line 278-279: In the sentence "Both isolates, which underwent WGS belonged to ST-383, a sequence type principally found in Europe as highlighted by the phylogenetic analysis," it may be clearer to state that the isolates belong to a sequence type predominantly found in Europe.

We have corrected the sentence

 Methodology:

 Section 4.1 Isolates and section 4.2 Antimicrobial susceptibility: The authors provide a clear and detailed description of the isolates and antimicrobial susceptibility testing methods used in the study, which enhances the reproducibility and transparency of their results.

Thank you for your comment

We have further revised the English language

Reviewer 3 Report

This study was aimed to evaluate cefiderocol in vitro-susceptibility of different multi-72 drug resistant Gram-negative bacteria isolates responsible of different infections. Despite the important clinical implications of this study, the authors must address some areas of concern.

Areas of concern:

Abstract

 This section does not mention the objective of the study.

Line 12: check spacing between ‘‘agent’’ and ‘‘against’’

Lines 23 and 27: write in italics ‘’Klebsiella pneumoniae’’ and ‘‘Enterobacterales’’

Line 30: The list of keywords seems to be incomplete

Introduction

Line 57: Write ‘‘antimicrobial’’ instead of’’ antimicrobic’’

Materials and methods

If there were no ethical considerations to address in this study, the authors should precise that the bacterial organisms were isolated from clinical specimen previously used for the routine work at the health facilities.

Lines 296-300: Put all the scientific names in italics.

Lines 326-349: specify the algorithms used for the different bioinformatics analyses.

References

Check references 2, 21,25,27,33,35 and 43 for consistency.

Author Response

Answers to reviewer 3

Abstract

 This section does not mention the objective of the study.

According to the reviewer’ s suggestion, we have added the aims of the study in the Abstract section

Line 12: check spacing between ‘‘agent’’ and ‘‘against’’

Thank you, we have added the space

Lines 23 and 27: write in italics ‘’Klebsiella pneumoniae’’ and ‘‘Enterobacterales’’

The mistakes have been corrected

Line 30: The list of keywords seems to be incomplete

We have added other keywords

Introduction

Line 57: Write ‘‘antimicrobial’’ instead of’’ antimicrobic’’ o

Thank you, the mistake has been corrected

Materials and methods

If there were no ethical considerations to address in this study, the authors should precise that the bacterial organisms were isolated from clinical specimen previously used for the routine work at the health facilities.

Thank you, we have added sentences about this point in the Notes

Lines 296-300: Put all the scientific names in italics.

The mistakes have been corrected

Lines 326-349: specify the algorithms used for the different bioinformatics analyses.

The different bioinformatic analysis were performed submitting the assembled contigs of each genome to the online available platforms listed in this section of Materials and Methods, therefore we are not able to describe the employed algorithms.

References

 Check references 2, 21,25,27,33,35 and 43 for consistency.

We have checked all these references.

Round 2

Reviewer 2 Report

I have reviewed the paper titled "In vitro activity of cefiderocol on multiresistant bacterial strains and genomic analysis of two cefiderocol resistant strains" and I am pleased to report that the authors have addressed all the concerns and suggestions I have made.